# FLiPS: Few-Shot Fingerprinting of LLMs via Pseudorandom Sequences

## Abstract

Identifying online Large Language Models (LLMs) via black-box queries, or fingerprinting, is now an active research problem. The state-of-the-art schemes require substantial amounts of queries to a model for building their fingerprints, often implying having it at hand. This precludes a swift fingerprinting of new models or variants freshly deployed online. In this paper, we propose FLiPS, a principled approach to LLM fingerprinting, which enables building a fingerprint using a trace number of queries (which we term few-shot). FLiPS exploits bias discrepancies in the generation of random binary sequences by LLMs for model identification. It employs the classical NIST cryptographic test suite to detect salient and interpretable differences in LLM outputs. We demonstrate that FLiPS achieves nearly 99% accuracy on a pool of 35 LLMs using as few as 40 queries to establish the fingerprint and 8 for its later identification. Furthermore, we propose an open-set environment where some models are unseen and must be labeled as such, and achieve 92.5% accuracy (with 67.6% on unseen models). This demonstrates that FLiPS achieves the novel task of the swift few-shot integration of new models in its operation.

## 1 Introduction

Identifying, i.e. *fingerprinting*, machine learning models executed at third parties has gained traction recently Zhao et al. (2025b); Godinot et al. (2025); Maho et al. (2023). This primitive for instance permits tracking potential leaks, with simple queries to these models as a source for identification. In the case of Large Language Models (LLMs), some works focus on being able to retrieve a particular model in which a fingerprint was primarily inserted (requiring weights access) Xu et al. (2024); Nasery et al. (2025); Russinovich & Salem (2024) while some others only use inherent model properties (i.e., in a *forensic* way), allowing to identify a large number of models Pasquini et al. (2025); Kurian et al. (2025). In all of these work, one either requires weights access (rather than forensic approach) or a large number of collected samples to build the fingerprint. Collecting such a number of samples may 1) prevent the possibility of considering a swift integration of new LLMs. This is a clear impediment when considering the current release rate of new LLMs. Moreover, 2) excessive querying may trigger defense mechanisms based on rate limiting as pointed out in Zhao et al. (2025a). Table 2 summarizes this point, highlighting that no existing work copes with only a handful of queries for its training operation in the forensic setup. This paper thus proposes FLiPS, a novel fingerprinting scheme remedying this lack.

The LLMmap scheme for instance Pasquini et al. (2025) processes LLM outputs corresponding to crafted queries for their discriminative power. These outputs are then encoded by a pre-trained text embedder (BERT-like Transformer) and used as features for classifier training. Once trained, the classifier can accurately identify the source model using only a small set of outputs (typically eight). In such a setup, the weight is on the construction of the fingerprint, a process in which a classifier is trained with a significant number of output sequences, so that effective discrimination between models is possible.

Independently, and in the aim to better understand the inner workings of LLMs, other research domains focus on quantifying to what extent LLMs are capable of generating random sequences when prompted for it Harrison (2024); Hopkins & Renda (2023); Van Koevering & Kleinberg (2024); Coronado-Blázquez (2025). They conclude that while most of prompted models show capabilities

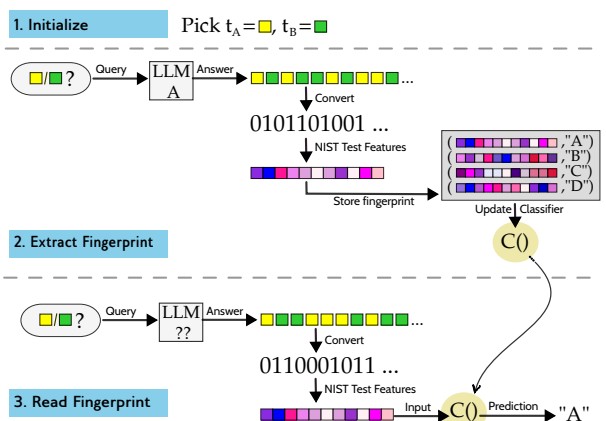

Figure 1: The FLIPS fingerprinting method. Two tokens $t_A$ and $t_B$ are selected (1) to query the model, which generates random binary sequences. Each sequence is then converted into a bit sequence to pass the NIST randomness test suite. The resulting test statistics serve as features to an XGBoost classifier (2). To read the fingerprint of a new black-box, new random sequences of the same two tokens are queried and fed to the trained classifier (3).

| Method | Few-Shot | Forensic |
|---|---|---|
| FLIPS (**Ours**) | ✓ (40) | ✓ |
| (Pasquini et al., 2025) | ✗ (∼1K) | ✓ |
| (Kurian et al., 2025) | ✗ (∼3.7K) | ✓ |
| (Xu et al., 2024) | **N/A** (≤60, requires insertion) | ✗ (fine-tuning) |
| (Nasery et al., 2025) | **N/A** (up to 24K, requires insertion) | ✗ (fine-tuning) |
| (Russinovich & Salem, 2024) | **N/A** (10–100, requires insertion) | ✗ (fine-tuning) |

Figure 2: Comparison of fingerprint construction schemes for LLMs, considering their requirements in terms of queries to build fingerprint (Few-Shot), and the LLM weights access (Forensic). The bottom three methods require first to have a complete access to the LLM, to insert information for later fingerprint reading in the wild.

in that task, there is important room for improvement. Traditionally, model discrepancies in tasks are often a fertile ground for their identification; be it simply on accuracy or classification failure patterns of simple input images Maho et al. (2023), or peculiar forms of semantic answers Pasquini et al. (2025) for instance. In that light, this work leverages the discriminative power of generated random sequences, for fingerpriting LLMs with FLIPS.

Relying on queries for random sequences has a native advantage over other fingerprinting techniques. Neural-network–based classifiers are often opaque, and this lack of transparency becomes even more pronounced when analyzing semantic sequences, such as phrases generated by LLMs Kim et al. (2020). A salient difference between LLM fingerprinting schemes such as LLMmap and random sequences (FLIPS) lies on outputs embedding. Where LLMmap uses a pretrained text embedder, FLIPS relies on the standard NIST Test Suite Bassham et al. (2010) for its classification. NIST is a celebrated cryptographic test suite for assessing the quality of pseudorandom sequences, and has been used for decades. FLIPS therefore relies on this set of explainable tests, assigning identified deviations from randomness to particular LLMs.

This paper makes the following contributions: 1) Our study motivates the development and demonstrates the possibility of a resource-efficient, few-shot fingerprinting technique for the seamless and stealthy integration of unseen LLMs in fingerprinting tasks. 2) We review and motivate why discrepancies in the generation of random sequences by LLMs are an attractive option for fingerprinting them. We propose to leverage the NIST suite for that task. For this, we introduce FLIPS, summarized in Figure 1, a three-step few-shot method to build and read the fingerprint of an LLM. Finally, 3) we evaluate on a pool of 35 modern LLMs, among which several state-of-the-art models, and quantify the benefits of FLIPS in a reproducible setup under two scenarios, closed and open-set.

The remainder of this paper is organized as follows. Section 2 presents the setting we are operating in. FLIPS is then presented in Section 3, before it is extensively evaluated in Section 4. We then discuss limitations and future work in Section 5, before related work is presented in Section 6. We make our reproducibility statement in Section 7 and then conclude.

## 2 PROBLEM SETTING

In a fingerprinting task, the attacker's objective is to infer a specific LLM version, placed for instance by its manufacturer, or by a malicious operator having leaked a protected model, on a remote server most frequently intermediated by an API. This paper operates with the following constraints in mind.

**Strict Black-box Access Setup.** n the black-box model, the attacker has query access to the model. They can only issue queries and retrieve the raw textual generation of the model, without any internal information (such as architecture, weights or even logits).

**Closed and Open-Set Scenarios.** Two scenarios are considered in this paper, *closed* set and *open* set. In a closed-set scenario, the black-box model belongs to a predefined set of known models. In an open-set scenario, the model may not belong to the known set, and should be classified as "unseen" rather than erroneously taken for a model in the closed-set.

**Few-Shot Fingerprint Construction.** A fingerprinting scheme must allow for the integration of unseen models by leveraging a small number of queries, for resource and computational efficiency.

**Excessive Query Detection.** Excessive querying patterns may trigger anomaly detection systems or rate-limiting mechanisms deployed by the target platform, as pointed out in Zhao et al. (2025a), potentially exposing the fingerprinting attempt and enabling defensive countermeasures. Thus, the fingerprinting scheme should minimize the number of queries when building the fingerprint of the source model and the number of reading queries issued to the target model. Few-shot training capability is therefore a crucial feature for a fingerprinting scheme, and to the best of our knowledge, this paper is the first work addressing it.

In that light, we address the problem of *Excessive Query Detection* and *Few-shot* capabilities with the proposal of a novel fingerprinting approach, adhering to the black-box setup and considering both closed and open-sets of LLMs.

## 3 THE FLIPS FINGERPRINTING SCHEME

### 3.1 PRELIMINARY DEFINITIONS

**Sequence Generation.** With FLIPS, we will query LLMs with a fixed prompt template $q_0$ (its exact formulation is provided in Appendix A), instructing the model to generate a random binary sequence. The baseline uses the symbols `0` and `1`, while alternative prompts substitute these with different tokens[1]. The set of all possible tokens is denoted by $\mathbf{T}$.

A model $m \in \mathcal{M}$ is defined such that that for a token pair $\mathcal{S} \in \mathbf{T}^2$, $m(\mathcal{S})$ is a random variable taking its values in $\mathcal{O} = \{0,1\}^*$ where $\{0,1\}^* \overset{\text{def}}{=} \bigcup_{K \in \mathbb{N}} \{0,1\}^K$. Sampling $m(\mathcal{S})$ corresponds to querying the LLM for a random sequence conditioned on the two tokens in $\mathcal{S}$. Note that $\mathcal{O} = \{0,1\}^*$ rather than $\{0,1\}^K$ for a fixed $K$, since different models may produce sequences of varying lengths and these variations are later exploited for fingerprinting. By defining $m(\mathcal{S})$ as a random variable, we encompass all stochastic elements of LLM generation, including sampling procedures (e.g., top-$k$/top-$p$) and hardware nondeterminism Atil et al. (2024).

**Sequence Embedding.** We define the embedding function

$$f : \mathcal{O} \longrightarrow \mathbb{R}^d$$
$$o \mapsto (f_1(o), \ldots, f_d(o)), \tag{1}$$

with $d \in \mathbb{N}^*$, where each component $f_i$ corresponds to a test statistic from the NIST Statistical Test Suite Bassham et al. (2010), except for one dimension reserved for the sequence length $|o|$. The specific tests employed are described in Appendix B.

**Classification Procedure.** We denote by $c_{\mathcal{S}, N_{\text{train}}}$ the classifier trained on datasets $\{\mathcal{D}_m\}_{m \in \mathcal{M}}$, where

$$\mathcal{D}_m = \{o_i^{(m)}\}_{i=1}^{N_{\text{train}}} \quad \text{with} \quad o_i^{(m)} \sim m(\mathcal{S}) \quad \text{i.i.d.} \tag{2}$$

---

[1] If $\mathcal{S} \neq$ `0`-`1`, the token sequence is converted into `0` and `1` by mapping first token to `0` and second to `1`.

and $N_{\text{train}} \in \mathbb{N}$ denotes the number of training samples. The classifier maps outputs $o \in \mathcal{O}$ to models $m \in \mathcal{M}$.

**Multi-Query Classification.** To do multi-query classification, i.e. using multiple queries for a single prediction, we aggregate the predictions made by $c_{\mathcal{S}, N_{\text{train}}}$ on the multiple queries into a single prediction using a soft-voting procedure (see Appendix C). Importantly, moving from single-query to $n$-query classification requires no additional training.

**Accuracy Metric.** We finally define the accuracy of the $n$-query classification task as:

$$acc(c_{\mathcal{S}, N_{\text{train}}}, n) = \mathbb{P}\Big(c_{\mathcal{S}, N_{\text{train}}}(O^{(m)}) = m\Big)_{m \in \mathcal{M}}, \ (O_i^{(m)})_{1 \leq i \leq n} \overset{\text{i.i.d}}{\sim} m(\mathcal{S}). \tag{3}$$

## 3.2 FLIPS Overview

FLIPS stands in those three steps (also depicted in Figure 1):

1. **Initialization.** Sample a random pair of tokens $(t_A, t_B) \in \mathbf{T}^2$.

2. **Fingerprint Training/Extraction.**

   (a) **Collecting Data.** Query $N_{\text{train}}$ times each LLM $m \in \mathcal{M}$ with $q_0(t_A, t_B)$, that is the query asking a random sequence of those two tokens (see Appendix A).

   (b) **Embedding** Convert the output sequences into bit sequences $o \in \{0, 1\}^*$ (cf Algo 5) and evaluate them with NIST (cf Appendix B) to get test statistics $\mathbf{x} = f(o) \in \mathbb{R}^d$.

   (c) **Classification Training** Train a classifier (XGBoost) on the test statistics $\mathbf{x}$, as features.

3. **Read a Fingerprint.** This classifier can fingerprint any black-box model from a new sample using $n$ queries, for any $n \in \mathbb{N}^*$.

With $\mathbf{T} = \bigcap_{m \in \mathcal{M}} \{t \in \mathbf{T}_m, |t| > 1, t \subset \Sigma_{\text{alnum}}\}$ where $\mathbf{T}_m$ is the set of all tokens[2] present in the vocabulary token of LLM $m$, when available (generally not the case for closed-weights models) and $\Sigma_{\text{alnum}}$ refers to alphanumeric characters.

## 3.3 Flexibility on Token Pairs

One can use any pair of tokens for binary sequence generation, since the mapping of the generated sequence to a bit-sequence representation is straightforward. We thereby set the maximum set of tokens shared by all the models we could (i.e. open-weights models used in the experiment). We explore that parameter space by investigating performance variability as a function of the sampled token pair. However, one has to keep the same token-pair at training and reading time (we investigated on cross token-pairs performances in Appendix D). Therefore, at this stage, this flexibility allows pivoting the token-pair, and consequently the classifier employed. We leave to future work the introduction of crossing token pairs within a single fingerprinting scheme.

## 3.4 NIST Test Statistics

As an illustration of the features FLIPS leverages, we here give a brief overview of the top-5 most empirically impactful NIST features, for the XGBoost classification procedure. The complete ranking is displayed in Figure 8 in the Appendix. Their description come from Bassham et al. (2010).

The Top-1 test, *Run*, measures the frequency of transitions between sequences of 0s and 1s. Top-2, *Monobit*, evaluates the proportion of zeros and ones across the entire sequence. Top-3 is not a NIST test but simply reports the output sequence length. Top-4, *Overlap 110*, counts the occurrences of the pattern `110` and finally, Top-5, *Longest-One Block*, measures the length of the longest run of consecutive 1s within a block (i.e., a subset) of the sequence.

Each test then compares its metric against the expected one under assumption of theoretical randomness.

---

[2]all *decoded* tokens

## 4 EXPERIMENTAL EVALUATION

We included 31 open-weights and 4 proprietary models (visible on Table 3) in our experimental evaluation. Some of these models are state-of-the-art for their manufacturers (e.g., gpt-5-nano, gemini-2.5-flash-lite or Llama-3.1-70B-Instruct). The generation configuration used can be found in Appendix E.

### 4.1 EXPERIMENTAL SETUP

**Token Pairs.** Our evaluation encompasses a total of 151 token pairs organized into three categories: **01** pairs: the baseline token pair 0-1; **Monochar**: 50 Mono-character pairs, sampled from a variation of $\mathbf{T}$ where tokens are a single character, and **FLiPS**: 100 pairs sampled from $\mathbf{T}^2$.

**LLM Sequences Gathering.** Details of how output sequences of LLMs were collected are provided in Appendix I. Also, a visualization of the generated sequence lengths is provided in Appendix J.

**Open vs Closed Set Setups.** In the open-set setting, the model set $\mathcal{M}$ is divided into two subsets: *Known* and *Unseen*. The objective is to construct efficient fingerprints for *Known* models while correctly labeling unseen models as *Unseen*. Our approach leverages the probability distribution over classes produced by the trained classifier. Indeed, for each input sample, the classifier assigns a probability to every class. In the closed-set setting, prediction corresponds to selecting the class with the highest probability. In contrast, for the open-set setting, a threshold is introduced: if the maximum predicted probability falls below this threshold, the sample is labeled as *Unseen*. Note that our thresholding method is tunable, allowing one to prioritize either *Unseen* or *Known* models.

Finally, the evaluation process is done within a 10-run cross-validation split with 40 samples for the test set and 40 for the training one. For closed-set approach, the crossing of splits is made over the samples while for the open-set, it is done over the samples but also the models. Indeed, within open-set approach, for each cross split, 5 models are isolated to constitute *Unseen* while the remaining 30 models make the *Known* pool. The evaluation procedure is further detailed in Appendix F.

### 4.2 FLiPS GENERAL PERFORMANCE

Experimental results show that FLiPS achieves almost 99% accuracy in closed-set setting and 92.5% in open-set with as few as 40 training samples per model and 8 queries at reading time, over a set of 35 LLMs including 4 proprietary models[3]. See Table 1 for an overview of the results, and Table 3 in Appendix G for a complete display of the results. In open-set setting, unseen models are retrieved with 67.6% accuracy, which is lower than for known models, however we propose a tunable parameter allowing to prioritize either unseen detection or known model identification. If scaling the number of training samples to 200 (in Closed-set), accuracy reaches 98% with only 3 queries at reading and 90% for a single query (see Figure 6).

The confusion matrix corresponding to the results of Table 3 (Closed-set) is in Figure 3. Interestingly, it explains the excellent results of FLiPS, as it displays the very high separability resulting from our relying on random sequences, of the models across both manufacturers and model variants. In particular, the closest proximity occurs between models in the same families, which is awaited: the two Phi-3-mini models and the two Phi-3-medium exhibit a proximity of 10% and 1% respectively.

### 4.3 FLiPS VS MONOCHAR AND 0-1 BASELINES

We now investigate whether certain token formats yield superior performance. We focus on how much the bits format 0-1 performs differently from the token pairs of $\mathbf{T}^2$ that are more conceptually

---

[3]Accuracy scores are computed using a two-stage averaging process. For each of the 151 token pairs, we first calculated the mean accuracy across its 10 cross-validation splits. Subsequently, to obtain category-level performance metrics for **01**, **Monochar**, and **FLiPS** token pairs, we computed the mean and standard deviation of these accuracy scores within each category. An exception is made for the baseline **01** as there is only one token pair in this group, so the standard deviation is shown over the 10 cross-validation splits.

| Model | Closed-set | Open-set |
|---|---|---|
| CohereForAI/c4ai-command-r-v01 | **99.02%** ($\pm0.69\%$) | **95.60%** ($\pm2.06\%$) |
| CohereLabs/aya-23-35B | **99.40%** ($\pm0.45\%$) | **96.80%** ($\pm1.71\%$) |
| HuggingFaceH4/zephyr-7b-beta | **98.32%** ($\pm1.29\%$) | **95.89%** ($\pm2.24\%$) |
| Qwen/Qwen2-72B-Instruct | **98.58%** ($\pm0.86\%$) | **91.84%** ($\pm2.84\%$) |
| abacusai/Smaug-Llama-3-70B-Instruct | **97.66%** ($\pm1.02\%$) | **86.78%** ($\pm3.70\%$) |
| anthropic/claude-3-haiku | **99.76%** ($\pm0.36\%$) | **98.32%** ($\pm1.42\%$) |
| deepseek/deepseek-chat | **98.22%** ($\pm0.91\%$) | **91.12%** ($\pm3.02\%$) |
| google/gemini-2.5-flash-lite | **98.34%** ($\pm1.79\%$) | **94.31%** ($\pm2.80\%$) |
| google/gemma-2-27b-it | **98.86%** ($\pm0.48\%$) | **88.44%** ($\pm3.24\%$) |
| meta-llama/Meta-Llama-3.1-70B-Instruct | **98.70%** ($\pm0.76\%$) | **93.07%** ($\pm2.42\%$) |
| microsoft/Phi-3-medium-128k-instruct | **96.52%** ($\pm1.58\%$) | **82.58%** ($\pm4.99\%$) |
| mistralai/Mistral-7B-Instruct-v0.3 | **98.32%** ($\pm1.14\%$) | **91.49%** ($\pm3.16\%$) |
| openai/gpt-4.1-nano | **98.26%** ($\pm0.97\%$) | **90.65%** ($\pm3.48\%$) |
| openai/gpt-5-nano | **99.74%** ($\pm0.26\%$) | **97.24%** ($\pm1.21\%$) |
| qwen/qwen3-next-80b-a3b-instruct | **99.34%** ($\pm0.54\%$) | **95.93%** ($\pm1.59\%$) |
| upstage/SOLAR-10.7B-Instruct-v1.0 | **97.24%** ($\pm1.40\%$) | **91.76%** ($\pm3.77\%$) |
| Unseen | — | **67.58%** ($\pm2.08\%$) |
| Average (on all 35 LLMs, see Table 3) | **98.72%** ($\pm0.77\%$) | **92.54%** ($\pm2.52\%$) |

Table 1: Short list of the 17 most advanced (model size by manufacturer) LLMs in our experiments (see Table 3 for the complete list of 35 LLMs). Fingerprinting accuracy of each model averaged over the 100 token pairs of FLIPS, evaluated under two scenarios: (i) closed-set, where the target model belongs to a known set of models, and (ii) open-set, where the target model may be unseen and must be correctly identified as such if unseen and as the right one if known. Performance is reported using 40 training samples to build the fingerprint and eight queries to read it.

exotic formats like *avid-ple* or *cards-quad* (see more examples of $\mathbf{T}^2$ in appendix H). Indeed, there is almost no chance that any sequence of these formats is present in a LLM's training set while it is almost sure that there are a lot of bit sequences as they represent binary data, making the comparison interesting. We also investigate the difference between tokens of a single character

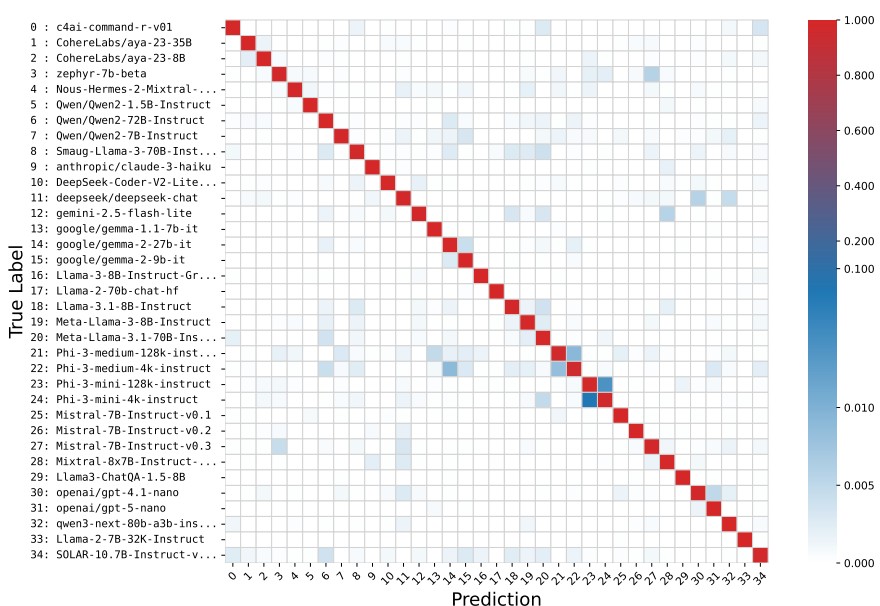

Figure 3: Confusion matrix in the closed-set setup. Averaged over the 100 token pairs of FLIPS. All rows sum to 1.0 (true labels are balanced over the classes).

(coined as Monochar) and tokens of **T** that are more than 1 character. Focus being on FLIPS, we halve the Monochar token pairs for computational efficiency.

Figure 4 and 5 compare the performances of FLIPS with the Monochar setting and `0-1` the baseline bit sequences. Overall FLIPS performs better, though very close to Monochar while the baseline pair `0-1` demonstrates notably inferior performance. This gap with the two other settings aligns with the conceptual distinction discussed above. However, no prior prediction was made regarding the direction of this gap.

Figure 4 shows the performances according to the number of queries used to read the fingerprint (the training samples remain 40). Performance improves substantially when increasing from 1 to 6 queries, but seem to plateau around 7–8 queries, indicating limited gains from additional queries. Figure 5 reflects the distribution of single-query accuracy achieved by individual token pairs across the three different settings. The exact bottom and top 5 token-pairs are available in Appendix H. Both FLIPS and Monochar categories demonstrate similar normal distributions, with FLIPS showing slightly superior mean performance.

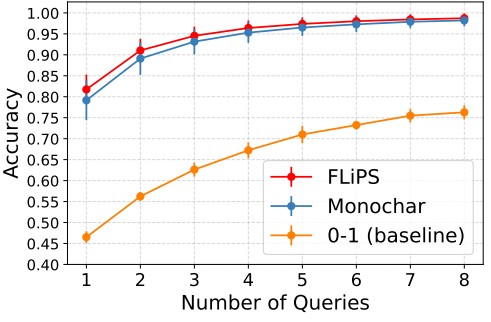

Figure 4: (Closed-set) Accuracy vs the number of queries used to predict the model. Mean and standard deviation are computed over all token pairs of the corresponding group of token pairs.

Figure 5: (Closed-set) Accuracy histograms achieved by each token pair, in a single-query scenario, and by the three different Token Pair competitors. See Appendix H for specific Bottom and Top-5 token pairs.

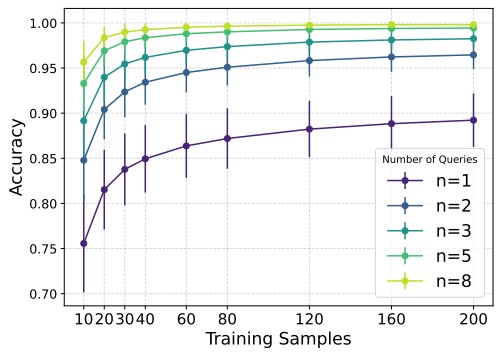
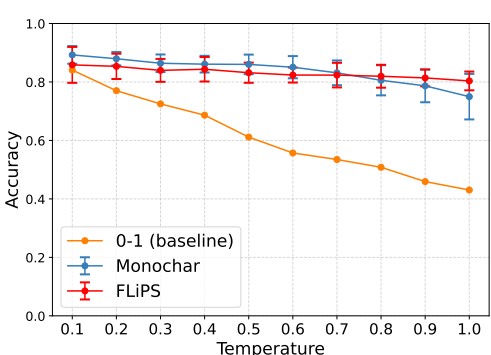

Figure 6: (Closed-set) Accuracy of FLIPS token pairs, as a function of different training samples ($N_{\text{train}}$). Each curve represents a different number of queries used for prediction. Mean and standard deviation are computed over the 100 token pairs.

Figure 7: (Closed-set) Temperature influence on accuracy, within single-query used for prediction. Mean and standard deviation are computed over all token pairs of the corresponding group of token pairs.

## 4.4 ABLATION STUDIES

We finally quantify the impact on FLIPS performances of the two following aspects.

**Influence of Training Samples**    Figure 6 illustrates model performance as a function of the number of available training samples (in closed-set setting). Performance appears to plateau after approximately 120 samples, particularly for $n \geq 3$, indicating that strong results can be achieved without requiring very large training sets (e.g., 98.49% accuracy with $n = 3$ and $N_{\text{train}} = 200$). Conversely, in very low-sample regimes ($N_{\text{train}} = 10$), accuracy ranges from still high values: 77.10% ($n = 1$) to 96.63% ($n = 8$).

**Impact of LLMs Temperature.**    Figure 7 shows how FLiPS is impacted by an increasing temperature, varying from 0.1 to 1.0. While the very purpose of the temperature parameter at a provider aims at introducing more randomness in the selection of next-tokens, we remark that it impacts negatively the `0-1` baseline; FLiPS and Monochar are also concerned, yet with a very small impact on both and in particular on FLiPS. Lowering the temperature parameter of LLM reduces the entropy of the output distribution, thereby narrowing the range of possible generations. The LLM then produces sequences that are more consistent across sequences. This increased regularity facilitates the classifier in learning and exploiting these biases.

## 5    DISCUSSION

Considering the many aspects at play in the fingerprinting of modern LLMs, we now discuss the most salient ones, for potential FLiPS limitations but also future work.

**Multi-Token Pair Classification for Enhanced Performance.**    Appendix D demonstrates that classifiers trained on token pair $\mathcal{S}_1$ fail to generalize to examples from a distinct token pair $\mathcal{S}_2$. This non-transferability property interestingly reveals that models exhibit distinct random biases when generating text conditioned on different token pairs. Since these bias variations encode complementary information signatures, employing multi-query classification across diverse token pairs could improve our fingerprinting scheme as future work.

**LLMs Accessing Tools.**    The advent of code execution capabilities in frontier LLMs, in particular with agents, might suggest that they can generate random outputs through computational means rather than relying on their inherent stochastic processes. However, current implementations maintain transparency and user control: web interfaces and APIs clearly indicate when code is being executed, preserving the foundational assumptions that make our fingerprinting approach viable.

In the same vein, the rise of agentic AI systems introduces new challenges for fingerprinting tasks. These systems remain nevertheless fundamentally enabled by LLMs, enabling fingerprinting through conventional text generation queries irrespective of their sophisticated scaffolding or multi-LLM configurations.

**LLM Capabilities Expansion.**    As LLMs advance, they may develop enhanced random generation capabilities without requiring code execution, reducing the discrepancies that FLiPS leverages. For instance, recent research has demonstrated that LLMs can internally embed deterministic algorithms within their weights, such as addition and multiplication schemes Kantamneni & Tegmark (2025); Maltoni & Ferrara (2024). However, this challenge can be addressed through progressively sophisticated randomness tasks: transitioning from binary to decimal integer generation, requiring adherence to specific probability distributions such as Poisson processes, or demanding complex random graph generation. Such escalating challenges could maintain FLiPS viability even against increasingly capable models, similar to the persistently useful fingerprinting schemes for image classifiers, still accurate despite the general increase in model accuracy Maho et al. (2023).

## 6    RELATED WORK

### 6.1    FINGERPRINTING IN THE CONTEXT OF LLMS

We start by clarifying the process of fingerprinting models with the related process of *watermarking* them. Fingerprinting employs a retrospective, forensic analysis of the object's existing characteristics to infer unique identifiers, without requiring any prior embedding. It relies on detecting inherent

patterns occurring in the system of interest Peng et al. (2022), Cao et al. (2021) and Pan et al. (2022). Most notably, the process of watermarking involves the intentional embedding of identifying features into the object to enable its identification, given knowledge of the embedding scheme. This proactive approach inserts markers to signal origin, authorship, or authenticity (See e.g., section 4 of Cox et al. (2006) or Kalker (2001))

**LLM specific methods**   In the context of LLMs, fingerprinting and watermarking techniques can be subdivided into two levels: models and generated content. We outline these levels to clarify the scope of what this paper does and does not address:

*Model Fingerprinting* (or Active Fingerprinting) uses carefully crafted inputs to elicit distinctive model behaviors. This approach is exemplified by our method and by Pasquini et al. (2025).

*Generated Content Fingerprinting* (or Passive Fingerprinting) identifies models by analyzing inherent features of generated content. Examples include methods based on logits statistics like perplexity Hans et al. (2024) and its curvature Mitchell et al. (2023), or log-rank information Su et al. (2023).

*Model Watermarking* embeds signals into model outputs for specific inputs, so that possession of the input and key to read the signal's output allows verification of the model's identity. Techniques include e.g. logit-distribution modifications on targeted inputs Xu et al. (2024); Nasery et al. (2025) and input–output hashing schemes Russinovich & Salem (2024). Although these works may use the term "model fingerprinting," we advocate for "model watermarking" to maintain alignment with prior literature and avoid confusion with the distinct concept of Model Fingerprinting as defined above.

*Generated Text Watermarking* Alters the LLM so it directly imprints signals into its generated outputs, typically via logit distribution biases Kirchenbauer et al. (2023) or more recently by selecting iteratively from multiple generation streams Giboulot & Furon (2024).

## 6.2    Biases in generating random numbers by humans and LLMs

Hopkins & Renda (2023) were the first to investigate the random number generation capabilities of LLMs, highlighting the challenges they face in this task. Interestingly, researchers in neurosensory sciences have shown that humans are identifiable according to their deviation from mathematical randomness when asked to generate some random numbers Schulz et al. (2021). Harrison (2024) remarks that ChatGPT-3.5 is better at the task than humans, yet lacks the "perfect evenness characteristic of pseudorandomly generated sequences." This is confirmed in earlier work Van Koevering & Kleinberg (2024). Finally, Coronado-Blázquez (2025) confirms that six models they tested have their own biases. These discrepancies in the task by LLMs motivated our work for an accurate fingerprint scheme, in which we are the first to consider the standard NIST suite for a principled analysis of pseudorandom outputs.

## 7    Reproducibility Statement

Experimental Section 4 and Appendices A, B, E, F and I provide sufficient information to reproduce the experiments. FLiPS code will be made available on GitHub should the paper be accepted.

## 8    Conclusion

As opposed to model-modifying watermarking schemes, fingerprinting schemes leverage particular traits in the outputs of the models they aim to identify. With FLiPS, we made the case for considering the discrepancy in the generation of random sequences by LLMs, for we demonstrated that they give excellent accuracy in the task, while permitting to train fingerprints with unseen models detection with a few tens of queries only. We believe this opens up an avenue for future works that are discriminative enough to seamlessly integrate the always evolving variety of modern LLMs.

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

APPENDIX

## A PROMPT $q_0(t_A, t_B)$

> **Random Binary Sequence Generation Prompt**
>
> Generate a sequence of 1000 symbols, each independently and uniformly sampled from the set $\{t_A, t_B\}$.
> Do not include any explanations, formatting, or additional text—only output the raw sequence.
> For reference, a sample sequence of 30 symbols is shown below:
> $t_A,t_A,t_B,t_A,t_A,t_A,t_A,t_A,t_B,t_A,t_A,t_A,t_A,t_A,t_A,t_A,t_B, t_A,t_B,t_B,t_A,t_A,t_B,t_B,t_B,t_A,t_A,t_B,t_A,t_A$
> Now, produce a sequence of 1000 such symbols:

$q_0(t_A, t_B)$, where $(t_A, t_B)$ are placeholders for token pairs such as those described in Appendix H.

## B NIST TESTS OF EXPERIMENTAL SETUP

The experiments were conducted using the following Python NIST implementation: `https://github.com/stevenang/randomness_testsuite`.

Table 2 presents all the NIST tests used to embed our sequences.

Although this paper does not focus on evaluating randomness quality, it should be noted that, due to the limited sequence lengths, the results obtained from the evaluated LLMs are insufficient to assess their randomness quality. Indeed NIST Test Suite is often calibrated for sequence of length $10^6$ and more.

Table 2: NIST tests used in experiment

| Test | Parameter | Value |
|------|-----------|-------|
| Block Frequency | Block size ($M$) | 30, 100 |
| Non-overlapping Template Matching | Pattern lengths; block size | 2, 3; 75 |
| Overlapping Template Matching | Pattern lengths; block size | 2, 3; 75 |
| Cumulative Sums | Digits | 1s, 2s |
| Monobit (Frequency) | — | — |
| Runs | — | — |
| Longest Run of Ones | — | — |
| Binary Matrix Rank | — | — |
| Spectral (DFT) | — | — |
| Approximate Entropy | — | — |
| Linear Complexity | — | — |
| Random Excursions | — | — |

For the Overlapping and Non-overlapping Template Matching tests, pattern lengths of 2 and 3 indicate that all aperiodic patterns of size 2 and 3 were used i.e. 8 patterns, yielding as many tests. Tests marked with "—" do not require specific parameters.

## C FORMAL DETAILS ON THE MULTI-QUERY CLASSIFICATION TASK

Formally, we defined a classifier $c$ as:

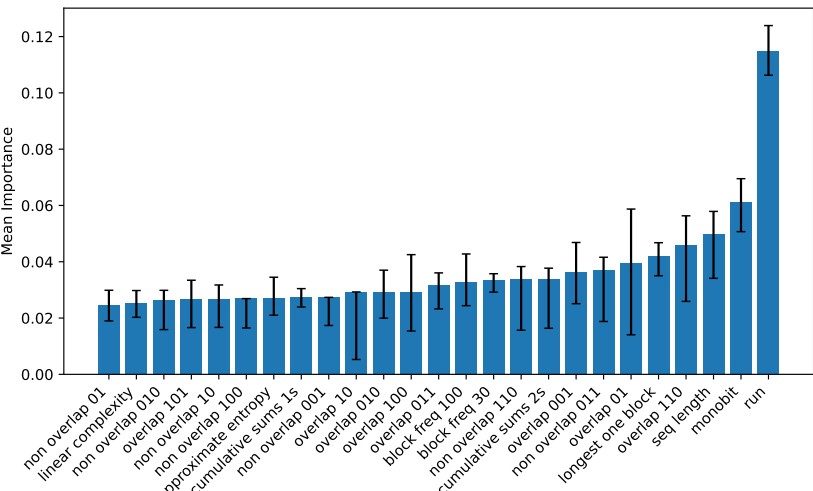

Figure 8: The top 25 most important features reported by the XGBoost classification. This ranking is made on averaged feature importance over all tested token pairs. The interquartile range is also reported as error bars.

$$c : \mathcal{O} \longrightarrow \mathcal{M} \tag{4}$$
$$o \mapsto \arg \max_{m \in \mathcal{M}} [g(f(o))]$$

where

$$g : \mathbb{R}^d \longrightarrow [0,1]^{\mathcal{M}} \tag{5}$$
$$\mathbf{x} \mapsto \big(P(m_1|\mathbf{x}), \ldots, P(m_{|\mathcal{M}|}|\mathbf{x})\big)$$

is a scoring function that maps feature representations to probability distributions over the model set $\mathcal{M}$. $P(m_i|\mathbf{x})$ represents the probability that model $m_i$ generated the output from which derives the feature vector $\mathbf{x}$.

We extend the single-query classifier to handle multiple outputs simultaneously. Each prediction is made by the single-query trained classifier, and are aggregated within a soft-voting procedure to make a final prediction. Soft-voting simply consists in averaging all probability distributions and then selecting the maximum value:

$$c_{\mathcal{S}} : \mathcal{O}^n \longrightarrow \mathcal{M} \tag{6}$$
$$(o_1, o_2, \ldots, o_n) \mapsto \arg \max_{m \in \mathcal{M}} \sum_{i=1}^{n} g_{\mathcal{S}}(f(o_i))_m$$

where $(o_i)_i$ are s.t. $\exists m \in \mathcal{M}, \forall i \in \{1, \ldots, n\}\, o_i \sim m(\mathcal{S})$, $g_{\mathcal{S}}$ is the scoring function trained on $\mathcal{S}$ and $g_{\mathcal{S}}(f(o_i))_m$ represents the probability, for $g_{\mathcal{S}}$, that output $o_i$ originates from model $m$ when queried with $\mathcal{S}$.

## D TOKEN-PAIR CROSSING

In this work, we convert generated sequences of token pairs into bit sequences and evaluate whether classifiers trained conditioned on one token pair $\mathcal{S}_{\text{train}}$ generalize to sequences conditioned on a different token pair $\mathcal{S}_{\text{test}}$. In the heatmap in Figure 9, each row shows the accuracies of a classifier trained on a single $\mathcal{S}_{\text{train}}$ when tested across many $\mathcal{S}_{\text{test}}$ (columns)[4]. The diagonal entries correspond to the within-pair case $\mathcal{S}_{\text{train}} = \mathcal{S}_{\text{test}}$ and therefore give the highest accuracies.

---

[4]To clarify, the number of classifier trainings equals the number of rows.

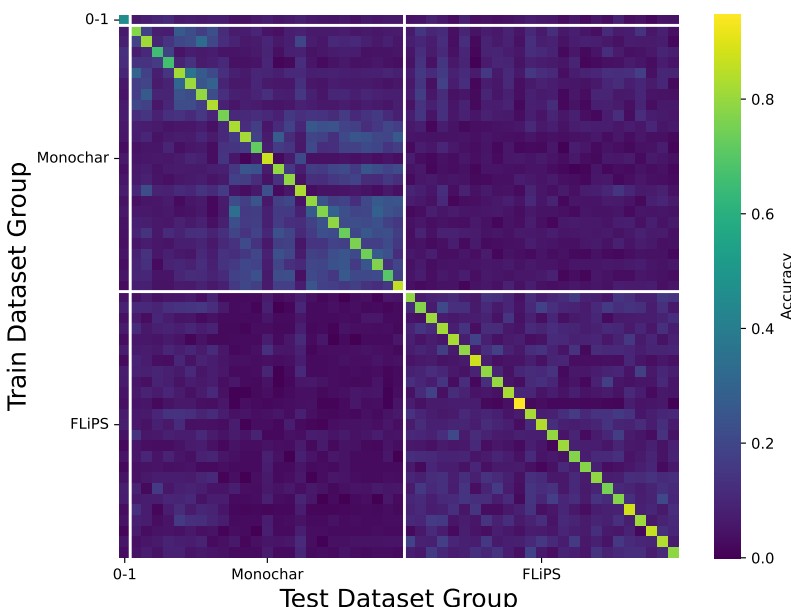

Figure 9: Classifier accuracy when trained on $\mathcal{S}_{\text{train}}$ (rows) and tested on $\mathcal{S}_{\text{test}}$ (columns). Diagonal entries are within-pair accuracies; off-diagonals show cross-pair transfer. (Experimental setup: $N_{\text{train}} = 40$, single-query $n = 1$; token pairs partitioned into three groups; subset sampled for display.)

Token pairs were partitioned into three groups; because there are up to 151 possible pairs, we randomly sampled a subset from each group for the heatmap. Overall, transferability across token pairs is negligible: off-diagonal accuracies are substantially lower than within-pair accuracies, indicating that the random biases of generated sequences depend strongly on the conditioning token pair.

There is modestly increased transferability when $\mathcal{S}_{\text{train}}$ and $\mathcal{S}_{\text{test}}$ belong to the same group, an effect more pronounced for the *Monochar* group.

Figure 10 summarizes these cross-pair results at the model level. For each model, we report the mean and standard deviation of accuracies across all $(\mathcal{S}_{\text{train}}, \mathcal{S}_{\text{test}})$ combinations restricted to each token-pair group (combinations across groups are excluded). Models exhibit similarly low cross-pair accuracy, except `Llama-2-70b-chat-hf`, which attains a higher mean (around 30%) with large variance, indicating that a few specific $(\mathcal{S}_{\text{train}}, \mathcal{S}_{\text{test}})$ pairs transfer substantially better for that model.

Consequently, if sequences generated from different token pairs encode complementary information, classification performance might improve by combining pairs, thereby increasing discriminative power and enhancing fingerprinting efficiency. However, mixing pairs while keeping the total number of training samples fixed reduces the number of samples *per* token pair, potentially leading to pair-specific underfitting. This trade-off highlights a direction for future work.

## E   GENERATION PARAMETERS

The target LLM operates under standard default sampling parameters, and without any fine-tuning operations.

**Default Parameters.** We adopt the default generation configuration recommended by Hugging Face, without any fine-tuning operations. We set the maximum number of generated tokens to 750, as this limit provides a relatively low computational cost while ensuring compatibility with platforms that impose stricter token constraints. This setup uses sampling with temperature $T = 1.0$, top-

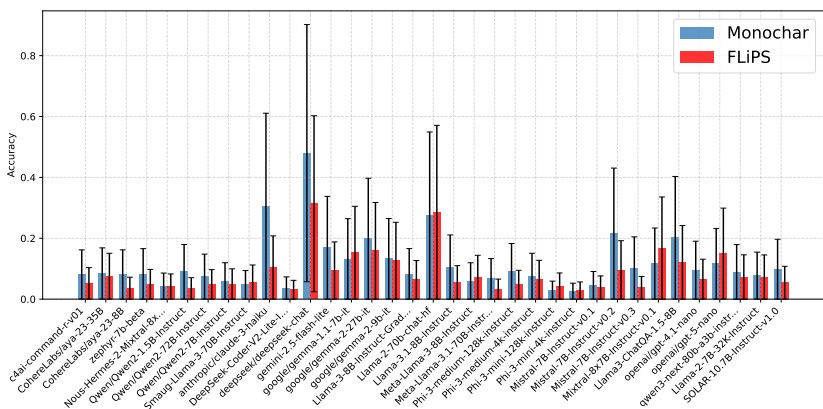

Figure 10: Per-model mean and standard deviation of cross-pair accuracies computed over $(\mathcal{S}_{\text{train}}, \mathcal{S}_{\text{test}})$ combinations within each token-pair group (same combinations as the upper left and lower right squares of Figure 9). (Experimental setup: $N_{\text{train}} = 20$, single-query $n = 1$.)

$k = 50$, top-$p = 1.0$, and a repetition penalty of 1.0. Only the $q_0$ prompt (Appendix A) is used, without system prompts or other additional components.

**Open-weights LLMs.** All open-weight models[5] are loaded in `bfloat16` precision and executed using the vLLM library Kwon et al. (2023) and a100 GPUs.

**Closed-weights LLMs.** All closed-weight models are accessed through the OpenRouter API with `reasoning=minimal`, since full reasoning suppression is not always available (for instance with the GPT-o series).

## F EVALUATION PROCEDURE

This section describes the detailed evaluation procedure for both closed-set and open-set scenarios. Algorithm 4 shows that we evaluate each token pair $\mathcal{S}$ independently.

**Closed-set.** The closed-set evaluation for a token pair $\mathcal{S}$ is summarized in Algorithm 2. First, we collect embedded output sequences from the LLM using FLiPS-Sampling (Algorithm 4). Next, we perform repeated cross-validation: on each split, we train a downstream classifier on the training fold and evaluate it on the test fold. Evaluation is performed in an $n$-queries regime for several values of $n$, yielding accuracies as a function of the number of queries. As described in Section 3.1, computing performance for different $n$ does not require retraining: predictions for multiple queries are aggregated via soft voting.

**Open-set.** The open-set evaluation is described in Algorithm 3. A subset of models in $\mathcal{M}$ is treated as *Known* (to be identified exactly) while the remaining models are treated as *Unseen* (to be labeled as such). Lines 3–5 collect embedded outputs for the `TrainKnown`, `TestKnown` and `TestUnseen` sets. We train a classifier on `TrainKnown` and then classify samples from both `TestKnown` and `TestUnseen`. In order to decide whether a sample is *Unseen* or not, we choose to leverage the probability distribution given by the classifier, and selecting a threshold below which the sample is predicted as unseen. Note that we omit to specify that accuracy is collected for each $n$-queries as in Closed-set, for clarity sake.

**Thresholding method.** Algorithm 6 outlines the procedure for constructing the decision threshold. In principle, one could fix a single threshold in advance using a dedicated (and possibly costly) procedure. However, in practice, it is more effective to determine the threshold dynamically from the available samples.

---

[5]except for Qwen3-80b that was also called with OpenRouter.

The dynamic approach replicates the open-set setting within the pool of *Known* models: we artificially separate them into *Known* and *Unseen* subsets to approximate real evaluation conditions. The objective is to identify a threshold that performs well in this replicated environment and then apply it to the actual task, hopefully generalizing well.

Concretely, we store the maximum predicted probability (i.e., the classifier's confidence in its assigned label) for each evaluated sample. These values form two distributions: one from correctly predicted Known samples and another from pseudo-unseen samples. Denote

$$\mathcal{P}_{\mathrm{K}} = \{p_{\max}^{(i)}\}_{i \in \text{Known, correctly predicted}}, \qquad \mathcal{P}_{\mathrm{U}} = \{p_{\max}^{(j)}\}_{j \in \text{pseudo-Unseen}}.$$

Three natural strategies for selecting the threshold $t$ are:

1. **Prioritize Known accuracy (our choice).** Choose $t$ as the $\alpha$-quantile of $\mathcal{P}_{\mathrm{K}}$, so that at most a fraction $\alpha$ of correctly predicted Known samples fall below $t$. This directly controls the tolerated fraction of Known samples that are labeled Unseen.

2. **Prioritize Unseen detection.** Choose $t$ to control the false negative rate on $\mathcal{P}_{\mathrm{U}}$ (for example, set $t$ so that a desired fraction of pseudo-Unseen samples fall below $t$).

3. **Optimize a global criterion.** Select $t$ to maximize an ROC-derived operating point, F1 score, or other combined metric computed from $\mathcal{P}_{\mathrm{K}}$ and $\mathcal{P}_{\mathrm{U}}$.

Figure 11 (a) visualizes the empirical distributions of $\mathcal{P}_{\mathrm{K}}$ (blue) and $\mathcal{P}_{\mathrm{U}}$ (red). Because the distributions overlap, threshold selection entails a trade-off between incorrectly rejecting Known samples and failing to detect Unseen samples; the strategies above make that trade-off explicit. Figure 11 (b) shows that thresholds vary widely, confirming the benefit of dynamic over static selection.

---

**Algorithm 1** FLIPS-Evaluation

---

**Require:** Number of token pairs $J$, model set $\mathcal{M}$, number of samples $N$, number of splits $N_{\text{split}}$, number of train/test samples $N_{\text{train}}, N_{\text{test}}$, maximum number of queries $N_{\text{queries}}$

1: Sample $(\mathcal{S}_j)_{j=1}^{J} \overset{\text{i.i.d.}}{\sim} \mathbf{T}^2$ {Draw $J$ token pairs}
2: Accuracy $\leftarrow \{\}$
3: **for** $j = 1$ to $J$ **do**
4:    **if** Closed-set **then**
5:       $Acc \leftarrow$ Closed-setEval$(\mathcal{S}_j)$ {Call Algorithm 2}
6:    **else**
7:       $Acc \leftarrow$ Open-setEval$(\mathcal{S}_j)$ {Call Algorithm 3}
8:    **end if**
9:    Accuracy$[\mathcal{S}_j] \leftarrow Acc$
10: **end for**
11: **return** Accuracies

---

**Algorithm 2** Closed-setEval (Evaluation of $\mathcal{S}$)

---

**Require:** model set $\mathcal{M}$, token pair $\mathcal{S}$, number of samples $N$, number of splits $N_{\text{split}}$, number of train/test samples $N_{\text{train}}, N_{\text{test}}$, maximum number of queries $N_{\text{queries}}$

1: Accuracies $\leftarrow \{\}$
2: Samples $\leftarrow$ FLIPS-Sampling$(\mathcal{S}, \mathcal{M}, N)$ {Call Algorithm 4}
3: **for all** $(\text{TrainSplit}, \text{TestSplit}) \in \text{CrossSplit}(\text{Samples}, N_{\text{split}}, N_{\text{train}}, N_{\text{test}})$ **do**
4:    $c_{\mathcal{S}, N_{\text{train}}} \leftarrow$ Train(TrainSplit, XGBoost)
5:    **for** $n = 1$ to $N_{\text{queries}}$ **do**
6:       Accuracies$[n] \leftarrow$ Accuracies$[n] \cup \{acc(c_{\mathcal{S}, N_{\text{train}}}, \text{TestSplit}n)\}$
7:    **end for**
8: **end for**
9: **return** Accuracies

---

## G COMPLETE RESULTS

The full results of the evaluation of FLIPS over the 35 LLMs are displayed in Table 3.

---

**Algorithm 3** Open-setEval (Evaluation of $\mathcal{S}$)

---

**Require:** Model set $\mathcal{M}$, maximum number of training samples per model $N_{\text{train}}$, number of test samples per model $N_{\text{test}}$, token pair $\mathcal{S}$
1: Initialize Acc $\leftarrow$ []
2: **for** each $(\mathcal{M}_{\text{Known}}, \mathcal{M}_{\text{Unseen}})$ in CrossSplit($\mathcal{M}$) **do**
3:     TrainKnown $\leftarrow$ FLiPS-Sampling($\mathcal{S}, \mathcal{M}_{\text{Known}}, N_{\text{train}}$) {Call Algorithm 4}
4:     TestKnown $\leftarrow$ FLiPS-Sampling($\mathcal{S}, \mathcal{M}_{\text{Known}}, N_{\text{test}}$)
5:     TestUnseen $\leftarrow$ FLiPS-Sampling($\mathcal{S}, \mathcal{M}_{\text{Unseen}}, N_{\text{test}}$)
6:     Threshold $\leftarrow$ BuildThreshold(TrainKnown, $\mathcal{M}_{\text{Known}}, \mathcal{S}$) {Call Algorithm 6}
7:     $c \leftarrow$ Train(TrainKnown, XGBoost) {Classifier training}
8:     Acc $\leftarrow$ Acc $\cup$ acc($c$, TestKnown $\cup$ TestUnseen, Threshold) {acc($\cdot$) computes closed-set accuracy on known models plus correct rejection rate on unseen models.}
9: **end for**
10: **return** Acc

---

**Algorithm 4** FLiPS-Sampling

---

**Require:** A Token Pair $\mathcal{S}$, model set $\mathcal{M}$, number of samples $N$
1: Initialize Samples $\leftarrow \{\}$
2: **for** each model $m \in \mathcal{M}$ **do**
3:     Initialize FeatureVectors $\leftarrow$ []
4:     **for** $i = 1$ to $N$ **do**
5:         Generate raw_output $\sim m(q_0(\mathcal{S}))$ {Generate sequence using model $m$}
6:         $o \leftarrow$ ExtractBit(raw_output) {Convert to bit sequence using Algorithm 5}
7:         $\mathbf{x} \leftarrow f(o)$ {Extract NIST feature vector}
8:         Append $\mathbf{x}$ to FeatureVectors
9:     **end for**
10:    Samples[$m$] $\leftarrow$ FeatureVectors
11: **end for**
12: **return** Samples

---

**Algorithm 5** ExtractBits

---

**Require:** string ans; binary_pair $(t_A, t_B)$
**Ensure:** bit-string $o \in \{0,1\}^*$
1: $m[t_A] \leftarrow$ "0", $m[t_B] \leftarrow$ "1"
2: $i \leftarrow 0$, $o \leftarrow$ ""
3: **while** $i < |\text{ans}|$ **do**
4:     **for** each item in $(t_A, t_B)$ **do**
5:         **if** ans$[i : i + |\text{item}|]$ == item **then**
6:             $o$ += $m[\text{item}]$; $i$ += $|\text{item}|$
7:             **break**
8:         **end if**
9:     **end for**
10:    **if** no match **then**
11:        $i$ += 1
12:    **end if**
13: **end while**
14: **return** $o$

---

---

**Algorithm 6** BuildThreshold

---

1: **Require:** Training set TrainSet, model set $\mathcal{M}$, token pair $\mathcal{S}$
2: Initialize MaxProbas $\leftarrow$ {Known : {}, Unseen : {}}
3: {Replicating open-set environment by creating pseudo-*Unseen* models, for threshold estimation.}
4: **for** each $(\mathcal{M}_{\text{Known}}, \mathcal{M}_{\text{Unseen}})$ in CrossSplit$(\mathcal{M})$ **do**
5:    TrainKnown, TestKnown $\leftarrow$ TrainTestSplit(Filter(TrainSet, $\mathcal{M}_{\text{Known}}$)) {Split subset of TrainSet corresponding to models of $\mathcal{M}_{\text{Known}}$}
6:    TestUnseen $\leftarrow$ Filter(Train, $\mathcal{M}_{\text{Unseen}}$)
7:    $c \leftarrow$ Train(TrainKnown, XGBoost) {Classifier training}
8:    MaxProbas[Known] $\leftarrow$ MaxProbas[Known] $\cup$ GetMaxProbas($c$, TestKnown)
9:    MaxProbas[Unseen] $\leftarrow$ MaxProbas[Unseen] $\cup$ GetMaxProbas($c$, TestUnseen)
10: **end for**
11: **Return:** ExtractThreshold(MaxProbas)

---

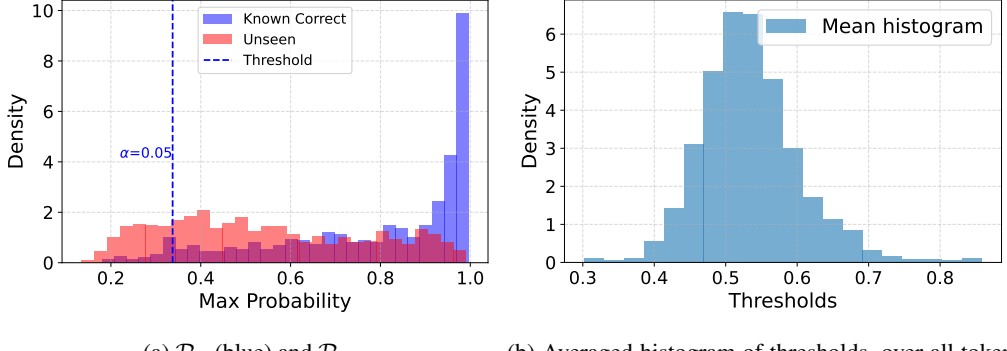

(a) $\mathcal{P}_{\text{K}}$ (blue) and $\mathcal{P}_{\text{U}}$ (red) with our chosen $\alpha = 0.05$.

(b) Averaged histogram of thresholds, over all token pairs.

Figure 11: (a) Illustration of an example of maximum probability distribution that needs to be separated by a threshold in Open-set. (b) Distribution of all the thresholds obtained with the thresholding procedure. Classification is made within a single-query here.

| Model | Closed-set | Open-set |
|---|---|---|
| CohereForAI/c4ai-command-r-v01 | **99.02%** $(\pm 0.69\%)$ | **95.60%** $(\pm 2.06\%)$ |
| CohereLabs/aya-23-35B | **99.40%** $(\pm 0.45\%)$ | **96.80%** $(\pm 1.71\%)$ |
| CohereLabs/aya-23-8B | **99.18%** $(\pm 0.50\%)$ | **97.08%** $(\pm 1.54\%)$ |
| HuggingFaceH4/zephyr-7b-beta | **98.32%** $(\pm 1.29\%)$ | **95.89%** $(\pm 2.24\%)$ |
| NousResearch/Nous-Hermes-2-Mixtral-8x7B-DPO | **98.46%** $(\pm 1.07\%)$ | **91.43%** $(\pm 3.27\%)$ |
| Qwen/Qwen2-1.5B-Instruct | **99.70%** $(\pm 0.27\%)$ | **98.60%** $(\pm 0.94\%)$ |
| Qwen/Qwen2-72B-Instruct | **98.58%** $(\pm 0.86\%)$ | **91.84%** $(\pm 2.84\%)$ |
| Qwen/Qwen2-7B-Instruct | **98.22%** $(\pm 0.97\%)$ | **87.80%** $(\pm 3.78\%)$ |
| abacusai/Smaug-Llama-3-70B-Instruct | **97.66%** $(\pm 1.02\%)$ | **86.78%** $(\pm 3.70\%)$ |
| anthropic/claude-3-haiku | **99.76%** $(\pm 0.36\%)$ | **98.32%** $(\pm 1.42\%)$ |
| deepseek-ai/DeepSeek-Coder-V2-Lite-Instruct | **99.02%** $(\pm 0.71\%)$ | **93.73%** $(\pm 2.52\%)$ |
| deepseek/deepseek-chat | **98.22%** $(\pm 0.91\%)$ | **91.12%** $(\pm 3.02\%)$ |
| google/gemini-2.5-flash-lite | **98.34%** $(\pm 1.79\%)$ | **94.31%** $(\pm 2.80\%)$ |
| google/gemma-1.1-7b-it | **99.92%** $(\pm 0.08\%)$ | **97.91%** $(\pm 1.01\%)$ |
| google/gemma-2-27b-it | **98.86%** $(\pm 0.48\%)$ | **88.44%** $(\pm 3.24\%)$ |
| google/gemma-2-9b-it | **99.62%** $(\pm 0.30\%)$ | **95.67%** $(\pm 2.05\%)$ |
| gradientai/Llama-3-8B-Instruct-Gradient-1048k | **99.70%** $(\pm 0.37\%)$ | **96.00%** $(\pm 1.59\%)$ |
| meta-llama/Llama-2-70b-chat-hf | **99.92%** $(\pm 0.09\%)$ | **96.86%** $(\pm 1.31\%)$ |
| meta-llama/Llama-3.1-8B-Instruct | **98.46%** $(\pm 0.90\%)$ | **92.24%** $(\pm 2.53\%)$ |
| meta-llama/Meta-Llama-3-8B-Instruct | **98.72%** $(\pm 0.72\%)$ | **91.90%** $(\pm 3.00\%)$ |
| meta-llama/Meta-Llama-3.1-70B-Instruct | **98.70%** $(\pm 0.76\%)$ | **93.07%** $(\pm 2.42\%)$ |
| microsoft/Phi-3-medium-128k-instruct | **96.52%** $(\pm 1.58\%)$ | **82.58%** $(\pm 4.99\%)$ |
| microsoft/Phi-3-medium-4k-instruct | **95.78%** $(\pm 1.99\%)$ | **82.82%** $(\pm 4.76\%)$ |
| microsoft/Phi-3-mini-128k-instruct | **97.66%** $(\pm 1.30\%)$ | **89.38%** $(\pm 3.65\%)$ |
| microsoft/Phi-3-mini-4k-instruct | **96.56%** $(\pm 1.75\%)$ | **89.36%** $(\pm 4.19\%)$ |
| mistralai/Mistral-7B-Instruct-v0.1 | **99.56%** $(\pm 0.46\%)$ | **95.75%** $(\pm 2.30\%)$ |
| mistralai/Mistral-7B-Instruct-v0.2 | **99.50%** $(\pm 0.34\%)$ | **95.67%** $(\pm 1.64\%)$ |
| mistralai/Mistral-7B-Instruct-v0.3 | **98.32%** $(\pm 1.14\%)$ | **91.49%** $(\pm 3.16\%)$ |
| mistralai/Mixtral-8x7B-Instruct-v0.1 | **99.12%** $(\pm 0.62\%)$ | **95.48%** $(\pm 2.17\%)$ |
| nvidia/Llama3-ChatQA-1.5-8B | **99.90%** $(\pm 0.10\%)$ | **96.52%** $(\pm 1.45\%)$ |
| openai/gpt-4.1-nano | **98.26%** $(\pm 0.97\%)$ | **90.65%** $(\pm 3.48\%)$ |
| openai/gpt-5-nano | **99.74%** $(\pm 0.26\%)$ | **97.24%** $(\pm 1.21\%)$ |
| qwen/qwen3-next-80b-a3b-instruct | **99.34%** $(\pm 0.54\%)$ | **95.93%** $(\pm 1.59\%)$ |
| togethercomputer/Llama-2-7B-32K-Instruct | **99.96%** $(\pm 0.05\%)$ | **97.75%** $(\pm 1.33\%)$ |
| upstage/SOLAR-10.7B-Instruct-v1.0 | **97.24%** $(\pm 1.40\%)$ | **91.76%** $(\pm 3.77\%)$ |
| Unseen | — | **67.58%** $(\pm 2.08\%)$ |
| Average | **98.72%** $(\pm 0.77\%)$ | **92.54%** $(\pm 2.52\%)$ |

Table 3: Fingerprinting accuracy of each model averaged over the 100 token pairs of FLIPS, evaluated under two scenarios: (i) closed-set, where the target model belongs to a known set of models, and (ii) open-set, where the target model may be unseen and must be correctly identified as such if unseen and as the right one if known. Performance is reported using 40 training samples to build the fingerprint and eight queries to read it.

# H  Token Pairs of T

| Token Pair FLiPS | Accuracy | Token Pair Monochar | Accuracy |
|---|---|---|---|
| **Top 5** | | | |
| ITION -- Fail | 0.94 | r -- 9 | 0.89 |
| ufact -- Eng | 0.89 | m -- 2 | 0.87 |
| acre -- Should | 0.89 | I -- 8 | 0.87 |
| Rich -- ody | 0.89 | X -- P | 0.86 |
| itudes -- oks | 0.89 | s -- 6 | 0.85 |
| **Bottom 5** | | | |
| um -- RO | 0.76 | S -- m | 0.73 |
| Cart -- ob | 0.76 | p -- r | 0.73 |
| dam -- trigger | 0.76 | W -- K | 0.69 |
| aly -- unc | 0.75 | 6 -- 9 | 0.68 |
| Pl -- go | 0.74 | 5 -- 2 | 0.67 |

Table 4: Top and Bottom 5 Token Pairs by Accuracy for FLiPS and Monochar relative to Figure 5.

# I  LLM Sequences Gathering Procedure

This section

Although the final number of training samples is 40, we collected additional samples to study the effect of sample size and token pairs. For each token pair and each LLM model, we then collected 240 samples, yielding a total of $240 \times 151 \times 35$ generated sequences.

Since LLMs produce sequences of varying length, we defined a minimum length threshold below which sequences were deemed too short to provide sufficient discriminative information. To reach the target of 240 valid samples, shorter sequences were discarded. For most models, very few sequences were discarded, so the reported number of training samples remains largely unaffected.

In further details, a sequence is considered *valid* if it contains at least 100 items, where an *item* is a token of the token-pair $\mathcal{S}$. During generation, we discard any sequence whose length is below this threshold and continue sampling until either (i) 240 valid sequences are obtained, or (ii) a per-experiment generation cap of 1000 sequences is reached.

Figure 12 reports, for each LLM, the average number of valid and discarded sequences across token pairs. Valid sequences are shown in blue and discarded sequences in red. Error bars indicate the standard deviation of the discarded-sequence counts across token pairs.

The five rightmost models, which predominantly produced short sequences, were excluded from the evaluation pool because they failed to generate a sufficient number of valid samples within the 1000-sequence cap. Specifically, all excluded models were either Base models (Mixtral-8x7B-v0.1, Qwen3-8B, Mistral-7B-v0.1, Orca-2-13B) or very small models (Qwen2-1.5B-Instruct, Gemma-2B-it). As a particularly illustrative example, Mixtral-8x7B-v0.1 yielded many discarded responses, whereas its instruct variant produced the fewest discarded responses overall. Therefore, their exclusion does not affect the conclusions regarding larger models.

For models that remained in the pool, some token pairs did not yield the full set of 240 distinct valid samples. In those cases, we applied *upsampling* (sampling with replacement) to reach the target of 240 samples; consequently, some sequences may appear multiple times across the training and test sets.

Finally, note that proprietary models are not included in the previous histogram, as far fewer sequences were collected for them in order to save credits. Only 80 samples were requested, and the number of discarded sequences was very low.

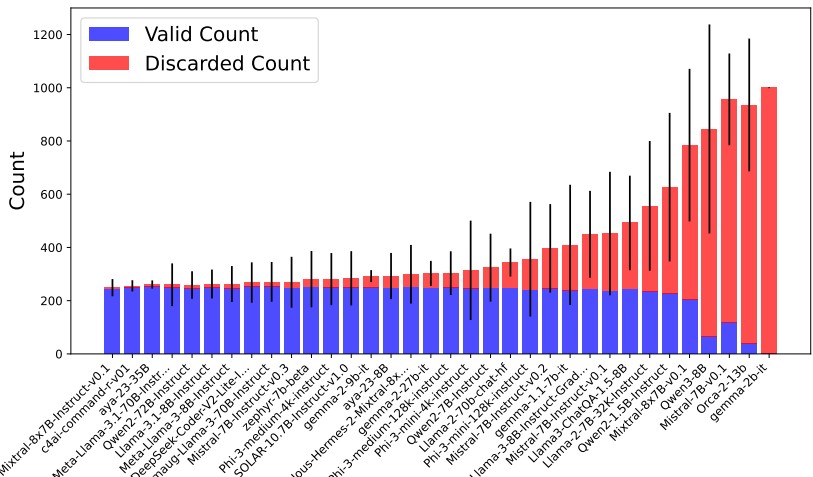

Figure 12: Mean number of valid and discarded generated sequences per LLM, averaged over token pairs. A sequence is valid if it contains $\geq 100$ items (items = tokens of the token-pair $\mathcal{S}$). Targets were 240 valid samples per (LLM, $\mathcal{S}$); generation stopped either when this target was reached or when a cap of 1000 total sequences was generated. Error bars show the standard deviation of the discarded counts across token pairs.

## J  GENERATED SEQUENCE LENGTHS

As described in Appendix I, in practice, LLMs do not generate sequences strictly within $\{0, 1\}^*$ or $\{t_A, t_B\}^*$ where $(t_A, t_B)$ is any token pair. Responses may contain extraneous content such as instruction acknowledgments, elaborations beyond requirements, or (mostly for base and discarded models) self-generated variations of the task. However, we extract only the binary elements from the output to construct the final sequence in $\{0, 1\}^*$ (with Algorithm 5).

Figure 13 presents the distribution of sequence lengths for each LLM across all available data. Since $q_0$ (cf. Appendix A) requires tokens to be comma-separated, it is expected that most LLMs produce sequences no longer than approximately MaxTokens$/2$, as each comma may consume one token. Nevertheless, some models exceed this limit and generate sequences up to MaxTokens. This behavior may arise either from variations in tokenization (where commas are embedded within tokens) or from non-compliance with the comma-separation instruction, given that Algorithm 5 does not explicitly enforce the presence of commas.

Finally, a few models produced (rarely) sequences exceeding MaxTokens, indicating that some tokens encapsulate multiple occurrences of the requested token.

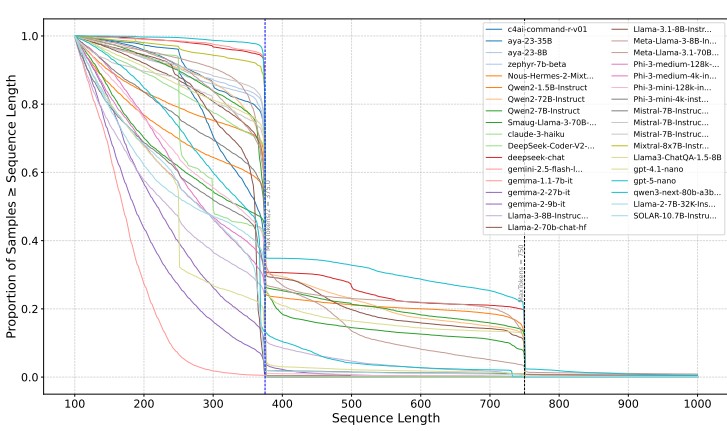

Figure 13: Average generated sequence lengths per model after bit extraction, computed over all token pairs $\mathcal{S}$ and the 240 collected samples. The blue and black dotted vertical lines indicate MaxTokens/2 and MaxTokens, respectively.

