# OpenReview forum: "FLiPS: Few-Shot Fingerprinting of LLMs via Pseudorandom Sequences"
_ICLR.cc/2026/Conference — Submitted to ICLR 2026_

### Official Review · Reviewer_pDkV · 2025-10-28

**Soundness:** 2
**Presentation:** 2
**Contribution:** 2
**Rating:** 4
**Confidence:** 4

**Summary:**

The paper proposes a method for fingerprinting large language models through few queries. The queries are designed to make an LLM output a random sequence of two tokens. A classifier is trained on NIST features extracted from this random sequence, and the classifier can achieve a high accuracy on new random sequences from known and unknown models.

**Strengths:**

1. The problem studied is important and practical
2. The method is simple and elegant, and achieves good results.
3. The experiments in the main paper are well executed to provide evidence for the method's effectiveness. I particularly liked the discussion around 0-1 not being an effective baseline, and I would have liked to see more such discussion around the proposed method's mechanisms.
4. I like the discussion around differentiating LLM fingerprinting from model watermarking (lines 442-453).

**Weaknesses:**

There are a few weaknesses of the work -

 ## Major Weaknesses
1. I am a bit skeptical that the paper's prompting strategy is good in the general blackbox (chat interface setting). I tried out the prompt in App A on some chat models (Gemini-2.5-Pro/Flash, GPT-5-Thinking, GPT-4o, GPT-4.1, Claude Sonnet-4/4.5) on the web interfaces with slightly problematic findings. I admit that this is much less scientific than the paper's experiments, having tried only once with one pair of tokens, and in the chat interface where the control over a lot of things is lot less. I observed that -

(a) GPT-5 thinking used the in-built python tool to generate code for random sequences

(b) Gemini-2.5-Pro started off fine, but quickly devolved into generating tokens other than those requested, before completely going off the rails and trying to emulate running a python script to do this task

(c) Gemini-2.5-Flash was fine for the most part, except that it did not terminate at all and actually hit the context limit

(d) GPT-4.1 and GPT-4o started off fine, before repeating one token indefinitely and hitting the context limit

(e) Claude-Sonnet-4.5 did not produce a response at all, triggering the safety filter

(f) Claude-Sonnet-4 was the only model which did not degenerate

This may be an artifact of using the web interface for chat, however, in an adversarial setting of model leakage it is possible that such an interface is the only way of interacting with the model. It could also be something more fundamental -- I believe that producing long, seemingly random pieces of text is not very stable for LLMs, and has been used as an attack to extract training data . E.g. Nasr et al asked a chat model to repeat a word (like poem) a hundred times and they noticed that the model soon devolved into outputting gibberish text from its training data. While this behaviour has been patched, the underlying issue might still be triggered by prompts such as the one in App A.

I am curious if the authors observed something like this in their experiments, and how they dealt with it.

2. **Lack of baseline comparisons** - Seems like Pasquini et al is the closest baseline to this method. How does it perform in the few (~100)-shot setting considered here?

3. What granularity does this method work on --- e.g can it detect if one LLM is a fine-tuned/quantized version of another? Further, did the experimental setup ensure that the model being served is the same for closed source APIs for train and inference (e.g. there are frequent reports of certain API providers serving quantized models at different times)?


### Subjective Critiques
I also have the following subjective critique of the paper -
1. **Contribution** - I am unsure who the intended audience of the paper is, or what the take-away from the paper should be.

(a) If it were intended for a general ML audience, is the take-away supposed to be that LLMs can generate random sequences in a unique but consistent way? If so, the reasons behind this phenomenon are left unexplained, and it seems that this phenomenon has been studied in prior work (cited in this paper as well). Concretely, could the authors conjecture why their method works?

(b) If the paper is intended for an LLM security audience, I feel the evaluation of the method is not adversarial or robust enough. This ties in to major weakness 3, i.e. the method should also be evaluated in settings where either system prompts are different, the model has been fine-tuned or quantized or hosted in a different inference provider etc.

2. **Claims of explainability** - The introduction stresses that the proposed method is more explainable than other methods, however I do not see direct evidence of this. What does explainability mean here? I believe that the argument is that using simpler classifiers leads to "more explainability". However I do not buy this argument fully, since the NIST features used do not seem particularly explainable or interpretable to me in terms of the broader task of distinguishing between LLMs.

## Minor

1. Some of the implementation details are unclear - E.g. line 1075 states upsampling was applied. Does this lead to train test leakage?
2. Nit - Improper citation style (please use \citep instead of \citet)

**Questions:**

Apart from the weaknesses above, I have the following questions-
1. How easy is it to add a new model? Do you need to only retrain the classifier?
2. Why does FLiPS limit the generation to selection from 2 tokens? Could the method be extended to use more tokens?
3. Do the authors have conjectures on why the classifier does not transfer across tokens?
4. How much variance do the different generated random sequences have for a single model?

---

### Official Review · Reviewer_9Tf5 · 2025-10-29

**Soundness:** 3
**Presentation:** 3
**Contribution:** 2
**Rating:** 2
**Confidence:** 3

**Summary:**

This paper introduces a new efficient fingerprinting technique, namely FLiPS. Intuitively, FLiPS probes the target models to generate a random binary sequence using two tokens; it provides the models with two tokens and requests them to generate a sequence of these tokens. FLiPS then converts the output into a binary sequence and extracts features using the NIST cryptographic test suite and other metrics (such as sequence length). These features are subsequently used to train a classifier for fingerprinting the models. The paper demonstrates strong results, achieving over 90% accuracy even when including new, unseen models (where the classifier only needs to identify them as unseen) using a limited budget of queries.

**Strengths:**

1. FLiPS shows strong performance even with unseen models.
2. It only requires a limited query budget, with results working for even a single query.
3. The interpretability of the features used by FLiPS.

**Weaknesses:**

FLiPS shows strong performance against different models in both open and closed source settings. However, I am not sure i understand the threat model as mentioned below.

   - It is not clear what the threat model for FLiPS is. I couldn't find any part discussing the threat model. Since FLiPS only needs API access, i believe more adversarial evaluation is needed. For example, using meta prompts to change the model's output—would this completely bypass FLiPS, since this would be an easy approach for an attacker/user who wants to hide their model?
   - Also, since FLiPS uses a very deterministic pattern (two words repeated multiple times), wouldn't this allow an attacker/user to filter/change this sequence to bypass FLiPS?
   - How does FLiPS deal with fine-tuning? If a model is fine-tuned, can it still be linked to the original model, or is it considered a new model? If all of these are out of scope, it would be really useful to clarify the use case.

**Questions:**

1. I am curious about the performance of exact matching (or allowing a small threshold) rather than using a classifier on top of the features. Would this be worse than using the classifier on top of features? In other words, how different are the features when using the same tokens for different generations?

2. In Line 325, it is mentioned halving the monochar token pairs for efficiency. What does this mean? Isn't a monochar a single token? Also, in this same section (4.1), I didn't really understand what the performance of the rare tokens is, or is that the performance of FLiPS?

3. In Figure 7, isn't the default result for temperature 1.0? Why does this figure show the maximum performance at ~80%? Even using a single query with 40 samples, the result is ~85%, as shown in Figure 6. I would suggest clarifying the settings for this plot.

4. It's a bit confusing that the paper classifies works such as Xu et al. (2024), Nasery et al. (2025), and Russinovich & Salem (2024) as watermarks, not fingerprints (which I agree are different), but then compares against them in Figure 2. Would also be helpful pointing to the prior works that established the differences between Watermarking vs. Fingerprinting

---

### Official Review · Reviewer_X6uG · 2025-10-30

**Soundness:** 3
**Presentation:** 3
**Contribution:** 3
**Rating:** 4
**Confidence:** 5

**Summary:**

This paper focuses on model copyright protection and proposes a fingerprinting method that leverages the inherent randomness characteristics of models. It constructs fingerprints based on these random features and trains an XGBoost classifier, achieving good performance in both open-set and closed-set classification scenarios. The paper also presents relatively sufficient experiments to evaluate the proposed method.

**Strengths:**

- The paper clearly identifies the shortcomings of existing approaches—namely, that intrusive fingerprints require weight access for embedding, and semantic fingerprints demand a large number of queries. Based on this, the motivation to utilize the inherent randomness of LLMs as a fingerprint is both reasonable and, in my opinion, quite promising and interesting.
- The main experiments are well-organized and complete. The tested models cover a wide range of types and sizes, and the paper conducts sufficient ablation studies on key parameters such as the number of training samples and verification queries.

**Weaknesses:**

Main issues

1. The paper does not adequately explore the resistance of its method to model-level attacks. In real-world scenarios, an adversary who steals a suspected model would typically perform light-weight retraining (e.g., SFT, DPO), pruning, model fusion, or perturb verification inputs. Many recently accepted papers have evaluated against such attacks.
2. The paper has not explored adaptive attacks. For instance, if the adversary perturbs inputs by randomly deleting 5% of the characters, would the detection still be reliable? The paper should at least include a discussion on this or consider more advanced adaptive attack strategies.
3. The necessity of the soft-voting procedure has not been explored—an ablation study is recommended.
4. Since only XGBoost is used as the classifier, it would be beneficial to include a comparison with an MLP classifier to demonstrate the discriminative potential of the proposed randomness features even under a weaker classifier.
5. There is no quantitative comparison with existing methods. I believe a comparison with LLMmap is essential for persuasiveness.

Typos and Minor Issues

- Line 113–116: “n the black-box model” should be “in the black-box model” (missing the letter ‘i’).
- The NIST cryptographic test lacks a brief explanation upon its first appearance; a short description or footnote is recommended.
- The term forensic setup is not common in this domain and may cause confusion (to my knowledge, no other works use it). Consider giving a detailed explanation when it first appears or reclassifying it under “semantic fingerprint” or “output-based fingerprint” as in prior works.
- In security research, the “PROBLEM SETTING” section might be better titled as “Threat Model” to more properly describe the adversary’s and defender’s capabilities.
- The definitions of T and T² are unclear at first mention; please refer readers to the examples in the appendix at Lines 142–143.
- The Related Work section is incomplete regarding model watermarking literature. It is recommended to consult the survey by Xu et al. [1] and cite more recent works such as TIBW [2] and CTCC [3].
- The structure of the Method section could be improved. Perhaps “FLIPS Overview” should appear at the beginning, followed by detailed descriptions of each component.

References:

[1] Copyright Protection for Large Language Models: A Survey of Methods, Challenges, and Trends
[2] TIBW: Task-Independent Backdoor Watermarking with Fine-Tuning Resilience for Pre-Trained Language Models
[3] CTCC: A Robust and Stealthy Fingerprinting Framework for Large Language Models via Cross-Turn Contextual Correlation Backdoor

**Questions:**

1. Regarding semantic-based fingerprinting (i.e., output-based fingerprinting), why is it necessary to model numerous existing models to predict whether a new model belongs to one of them or is unseen? For a model owner aiming to protect a particular model, isn’t it sufficient to perform binary classification (i.e., whether the suspect model belongs to this model or not)? Does this suggest a potential issue in the initial experimental setup? (I understand this work might follow LLMmap, but I hope the authors can clarify this point.)

2. How should one interpret the statements in the introduction: “prevent the possibility of considering a swift integration of new LLMs” and “the weight is on the construction of the fingerprint”?

3. Could the authors provide an example to clarify Line 213–214: “Each test then compares its metric against the expected one under assumption of theoretical randomness”?

4. How is the threshold between seen and unseen models determined?

5. Why does FLiPS outperform the 0–1 baseline? Can the authors provide a theoretical justification beyond empirical observations? The explanation that it arises from “conceptual distinction” may be overly intuitive.

6. Since models within the same family share more similar characteristics, does this affect the accuracy of the proposed method?

If the authors can include comparisons with LLMmap (most important), results under incremental training and input perturbation attacks, as well as results using MLP as a classifier, I would raise my score to 6–8.

---

### Official Review · Reviewer_NbXP · 2025-11-01

**Soundness:** 3
**Presentation:** 3
**Contribution:** 2
**Rating:** 2
**Confidence:** 4

**Summary:**

The paper proposed a method to test whether two models are related using a small number of (long generations). They key idea is to view the model as a psuedorandom number generator, and to measure the quality using features from the NIST cryptographic test suite. Once these features are obtained, an XGBoost classifier is trained on them to determine whether two models are related.

**Strengths:**

1. The objective of the method is sensible. For example, for models that are only accessible via a GUI chat interface, it may be impractical to generate thousands of responses needed by other fingerprinting methods.
2. The use of the NIST cryptographic test suite is clever and suits the setting and objectives well.

**Weaknesses:**

1. Does not examine robustness to common modifications made during LM deployment, including (for example): fine-tuning (adaptation), quantization, prompting (e.g. system prompts), etc. I have no reason to suspect this fingerprint is less robust than others in this aspect, but it's important to show that the fingerprint will persist in realistic scenarios.
2. In situations where the model provider controls the sampling (e.g. a chat app), the provider can easily detect the fingerprint by monitoring the token stream. In this case, the sampler can force true psuedorandom generation, continue the response with another model, or take any of a number of other interventions. Unless this can be addressed with more careful response design, this makes the fingerprint suitable only when the tester controls the sampling (e.g. if they are in possession of the weights) .
3. The usage of a trained classifier to determine model similarity does not give any guarantees regarding the false-positive or false-negative rates (unlike a hypothesis test). Typically, it is important to control the false-positive rate so one can be certain they are not claiming ownership of a model that is not theirs.

**Questions:**

1. For the reasons highlighted in weaknesses 2 and 3, the design of the scheme may not be well-suited for fingerprinting. However, it does fit the setting of model equality testing [1] quite well. Perhaps that may be a good application for this method?

[1] Gao, Irena, Percy Liang, and Carlos Guestrin. "Model Equality Testing: Which Model Is This API Serving?." arXiv preprint arXiv:2410.20247 (2024).

---

### Meta-Review · Area_Chair_YQ52 · 2026-01-01

**Summary:**

This paper proposes FLiPS, a few-shot fingerprinting method for black-box identification of large language models based on biases in pseudorandom sequence generation, using features derived from the NIST cryptographic test suite and a lightweight classifier. Reviewers broadly agree that the problem is important and timely, and that the approach is simple, query-efficient, and empirically effective across a diverse set of models, including open-set scenarios. However, reviewers also raised substantial concerns regarding the realism of the threat model, robustness to common deployment and adversarial modifications, reliance on a trained classifier without statistical guarantees, and the lack of comparisons with closely related baselines. Notably, the authors did not submit a rebuttal, and thus none of these concerns were addressed or clarified during the response phase. As a result, the evaluation necessarily reflects only the originally submitted manuscript and the reviewers’ initial assessments, which informed the final recommendation.

**Reviewer Concerns:**

Across reviews, there was general agreement that FLiPS is a promising and practically motivated approach to few-shot LLM fingerprinting, with strong empirical performance and a compelling reduction in query requirements. Reviewers appreciated the use of randomness-based features and the interpretability of NIST-style tests, as well as the relatively broad experimental coverage across models and settings.

At the same time, reviewers consistently raised several major concerns that remain fully outstanding due to the absence of an author rebuttal. In particular, the paper does not clearly specify or justify a realistic threat model for online LLM fingerprinting, especially in settings where the model provider controls sampling, system prompts, or post-processing, or where lightweight model modifications such as fine-tuning, quantization, pruning, or model fusion are applied. Robustness against adaptive or adversarial strategies, including input perturbations or output filtering, was not evaluated. Reviewers also noted the lack of quantitative comparisons with closely related model identification or fingerprinting methods, which limits the ability to assess the method’s relative advantages. Additionally, the reliance on a trained classifier without explicit control over false-positive or false-negative rates raises concerns about suitability for ownership claims or forensic use. Since no rebuttal was provided, these concerns could not be addressed, clarified, or mitigated, and therefore remained central to the final assessment.

**Reviewer Scores:**

Given that no rebuttal was submitted, reviewers did not have the opportunity to reconsider their assessments in light of clarifications, additional experiments, or scope adjustments. As a result, I do not expect reviewer scores would have meaningfully increased even under a full discussion period. Reviewers who were already moderately positive about the empirical results might have maintained their scores, but the major concerns regarding threat model realism, robustness, and evaluation completeness would likely have persisted. Overall, in the absence of author responses, the paper would plausibly remain in the same mixed to borderline-reject range rather than converging toward acceptance.

---

### Decision · Program_Chairs · 2026-01-26

Reject